# OpenReview forum: "A Closer Look at System Message Robustness"
_NeurIPS.cc/2024/Workshop/SafeGenAi — SafeGenAi Poster_

### Official Review · Reviewer_yUHh · 2024-10-08
**Review of A Closer Look at System Message Robustness**

**Rating:** 7
**Confidence:** 4

**Review:**

This paper explores the robustness of system messages in large language models (LLMs), specifically focusing on their susceptibility to adversarial attacks such as prompt injection. System messages are used to control the behavior of LLMs by specifying rules and instructions that should take precedence over user inputs. However, these instructions are often violated in practice, especially under adversarial conditions. To address this, the authors develop a new dataset (SUDO) and benchmarks (RULES, Gandalf, TensorTrust) to assess system message robustness, and fine-tune LLMs to improve adherence to system messages. Overall, I think this paper is of good practical use to the research community.
Pros:
(1)	The introduction of the SUDO dataset, which is constructed from real-world system prompts and includes both benign and adversarial user messages, is a significant contribution. The dataset is carefully designed to cover a wide range of scenarios, including tool-assisted LLMs (e.g., browsing, web search, etc.), making it highly applicable to practical use cases.
(2)	The authors provide a clear and well-structured evaluation framework with benchmarks such as RULES, Gandalf, and TensorTrust.
Some concerns:
(1)	While the paper provides strong benchmarks for system message adherence, it does not fully explore how well these methods generalize across different types of LLMs. I agree that Mistral-7B is a relevant model but it’s much better to show performance of models from different families (Llama etc.)
(2)	Although the SUDO dataset is a valuable contribution, its size (13.9K training samples and 3.8M tokens) is still relatively small. I think authors need to discuss the potential limitations more thoroughly.
(3)	I suggest authors adding a paragraph that addresses the trade-offs in deploying such robust models, such as potential increases in inference time, memory usage, or hardware costs. This would make the paper more complete from a practical standpoint.
(4)	Finally, what would be the potential adversarial bypasses? I would very much like to see some discussions on whether more sophisticated adversarial attacks could still exploit vulnerabilities.